# Charge Dirichlet Energy: Geometric Perspectives on Over-smoothing in Deep Graph Neural Networks

## Abstract

Over-smoothing is regarded as a key issue affecting the performance of deep Graph Neural Networks (GNNs). As the number of GNN layers increases, model performance degrades significantly, due to node embeddings converging into indistinguishable vectors. This phenomenon stems from the recursive aggregation of neighbor node representations, which impairs the distinguishability of node embeddings. From an energy perspective, this is associated with the convergence of node embeddings to a fixed point solution during the minimization of Dirichlet energy, hindering the model's ability to learn underlying geometric structures. While Graph Convolutional Networks (GCNs) have achieved success in modeling graph-structured data, there is still insufficient understanding of how the underlying geometry contributes to the trainability of deep GCNs. In this paper, we present a novel geometric perspective to understand the poor performance of deep GCNs during training, a method called Charge Dirichlet Energy (CDE-GNN). We argue that maintaining a healthy geometric structure can significantly enhance the trainability of GCNs and enable state-of-the-art performance, even in base GCN architectures. Subsequently, we analyze the importance and feasibility of learning geometric shapes, demonstrating the critical role of geometric information in training deep GNNs. Extensive empirical validation on multiple benchmark datasets shows that our method improves the geometric shape of deep base GCNs, significantly enhancing their performance and outperforming many state-of-the-art methods in competitive settings. Our contributions include not only a new approach to mitigating over-smoothing and over-compression but also comprehensive theoretical and empirical verification of the importance of geometric structures for the trainability of deep GNNs.

## 1 Introduction

GNNs have recently emerged as a hot topic in computer science and artificial intelligence Gori et al. (2005); Scarselli et al. (2008); Duvenaud et al. (2015); Hamilton et al. (2017); Xu et al. (2018a). GNNs have found widespread applications in fields such as computer vision and graphics Monti et al. (2017); Wang et al. (2018); Eliasof & Treister (2020), social network analysis Kipf & Welling (2016); Defferrard et al. (2016), and bioinformatics Jumper et al. (2021). Most GNNs adopt the *message passing* paradigm Gilmer et al. (2017), where learnable nonlinear functions propagate information across the graph Kipf & Welling (2017); Veličković et al. (2018). Specifically, information from neighboring nodes is iteratively aggregated and used to update central node representations, making GNNs well-suited for modeling complex relational structures (nodes and edges) in graph-structured data. Many real-world domains naturally exhibit graph structures, and tasks based on graph structures, such as social analysis Qiu et al. (2018), traffic forecasting Guo et al. (2019); Li et al. (2019b), biology Fout et al. (2017); Shang et al. (2019), recommendation systems Ying et al. (2018), and computer vision Zhao et al. (2019), are commonly modeled using GNNs.

However, GNNs generally follow a common message-passing paradigm Gilmer et al. (2017), which has significant limitations. These include restricted expressiveness Xu et al. (2019); Morris et al. (2019), over-compression Alon & Yahav (2020); Di Giovanni et al. (2023), and the inability to capture long-range dependencies Li et al. (2018). Additionally, the propagation operators in most

common architectures are constrained to be non-negative, leading to a smoothing effect in the propagation process, which can result in over-smoothing Li et al. (2018); Chen et al. (2020a). As more layers are stacked, node features become indistinguishable, and the performance of deep GNNs degrades significantly Zhao & Akoglu (2020b); Nt & Maehara (2019); Oono & Suzuki (2020); Cai & Wang (2020). This phenomenon corresponds to the excessive shrinkage of Euclidean distances between nodes, resulting in the loss of distinguishing information. Consequently, in practice, most tasks only require a few layers (two or three) Qu et al. (2019).

Recent studies have analyzed the training of GNNs from the perspective of Dirichlet energy Cai & Wang (2020); Zhou et al. (2021), showing that as the network depth increases, Dirichlet energy decays to zero, limiting the expressive power of GNNs.

In particular, during the process of over-smoothing, the node representations' feature magnitudes diminish Oono & Suzuki (2019), high-frequency features are filtered out, and low-frequency features are diffused into noise Wang & Leskovec (2020). Geometrically, the norms of node representations contract, converging toward a fixed point Gu et al. (2020); Liu et al. (2021); Chen et al. (2022); Liu et al. (2022). This leads to edge-space contraction, structural collapse, and the loss of geometric information. To increase model capacity, researchers have employed residual connections and initial connections Xu et al. (2018c); Li et al. (2019a); Chen et al. (2020b) to alleviate over-smoothing and improve model depth and capacity. However, model performance does not always improve with increased depth.

To address these challenges, we propose a geometry-driven framework that designs learnable propagation mechanisms based on a parameterized graph Laplacian operator. We define Hilbert spaces on both vertices and edges, leveraging Dirichlet energy defined on edge space to measure the smoothness on the graph. These parameterized methods provide flexibility in learning the geometric shapes of vertex and edge spaces from data. To prevent Dirichlet energy from collapsing to zero, we impose a minimum Dirichlet energy $\omega$ on node representations, effectively preventing unreasonable contraction of edge space and mitigating the homogeneity of node features. From the perspective of edge space, the learnable $\omega$ allows the operator to flexibly adjust the distances between nodes, avoiding distance collapse caused by over-smoothing and enhancing the robustness of node representations by preserving geometric structure.

We validate the effectiveness of our model and theoretical results on various benchmark datasets. Experimental results show that in most cases, our model outperforms both explicit and implicit GNN baselines in two types of tasks, demonstrating significant advantages in addressing the issues of over-smoothing and over-compression. Our main contributions are as follows:

- We propose a geometric framework based on a parameterized graph Laplacian operator, aimed at mitigating the problems of over-smoothing and over-compression in deep GNNs.

- We theoretically analyze the importance of geometric shape learning and its profound impact on the trainability of deep GNNs, rigorously proving the critical role of geometric information in enhancing overall model performance and effectiveness.

- Through comprehensive and extensive empirical validation on diverse benchmark datasets, we conclusively show that our innovative approach significantly improves the performance of deep GCNs across various real-world scenarios, consistently outperforming numerous state-of-the-art methods in comparative evaluations.

## 2 RELATED WORK

**Notation.** Consider an undirected graph $\mathcal{G} = (\mathcal{V}, \mathcal{E})$, where $\mathcal{V}$ consists of $n$ vertices and $\mathcal{E}$ contains $m$ edges. For each vertex $i$ in $\mathcal{G}$, its feature vector is denoted by $\mathbf{f}_i \in \mathbb{R}^c$, where $c$ represents the number of channels. The adjacency matrix is denoted as $\mathbf{A}$, where $\mathbf{A}_{ij} = 1$ if there is an edge $(i, j) \in \mathcal{E}$ and $\mathbf{A}_{ij} = 0$ otherwise. The degree matrix is denoted as $\mathbf{D}$, with diagonal elements $\mathbf{D}_{ii}$ representing the degree of vertex $i$. The graph Laplacian is defined as $\mathbf{L} = \mathbf{D} - \mathbf{A}$. For graphs with self-loops, we introduce the modified adjacency matrix $\tilde{\mathbf{A}}$ and the corresponding degree matrix $\tilde{\mathbf{D}}$. The symmetrically normalized Laplacian matrix is denoted as $\tilde{\mathbf{L}}^{\text{sym}} = \tilde{\mathbf{D}}^{-\frac{1}{2}} \tilde{\mathbf{A}} \tilde{\mathbf{D}}^{-\frac{1}{2}}$.

**Deep GNN Architectures.** To enhance the depth and performance of GNNs, various innovative and sophisticated architectures have been proposed, such as DeepGCN Li et al. (2019a), JK-Net Xu et al. (2018c), MixHop Abu-El-Haija et al. (2019), DAGNN Liu et al. (2020), EGNN Zhou et al. (2021), and GCNII Chen et al. (2020b). These carefully designed architectures introduce residual connections across layers or within a single layer, enabling more effective and efficient propagation of features in deep graph structures without relying on computationally expensive sampling methods.

**Over-smoothing in GNNs.** The phenomenon of over-smoothing was first highlighted in Li et al. (2018) and has since been extensively studied. Several strategies have been proposed to mitigate over-smoothing based on different approaches. For instance, DropEdge Rong et al. (2020), PairNorm Zhao & Akoglu (2020a), and EGNN Zhou et al. (2021) leverage data augmentation, normalization, and energy-based regularization, respectively, to alleviate over-smoothing. Additionally, Min et al. (2020) enhances GCNs by incorporating geometric scattering transforms and residual convolutions. GCNII Chen et al. (2020c) addresses over-smoothing by analyzing spectral smoothness and incorporating identity residual connections and deep weight decay, techniques that are also employed in EGNN Zhou et al. (2021).

**Definition 2.1** (Dirichlet Energy Cai & Wang (2020)). *Given the node embedding matrix at the $k$-th layer $X^{(k)} = [x_1^{(k)}, \cdots, x_n^{(k)}]^\top \in \mathbb{R}^{n \times d}$, the Dirichlet energy $E(X^{(k)})$ is defined as:*

$$E(X^{(k)}) = \frac{1}{2} \sum_{i,j} a_{ij} \left\| \frac{x_i^{(k)}}{\sqrt{1+d_i}} - \frac{x_j^{(k)}}{\sqrt{1+d_j}} \right\|_2^2, \tag{1}$$

*where $a_{ij}$ represents the edge weight between nodes $i$ and $j$, and $d_i$ is the degree of node $i$. The Dirichlet energy quantifies the smoothness of the embeddings by measuring the weighted distance between node pairs.*

## 3 OVER-SMOOTHING AND EDGE-SPACE COLLAPSE

### 3.1 NODE SPACE AND DIRICHLET ENERGY

**Definition 3.1** (Inner Product in Vertex Space). *Let $\mathcal{G} = (\mathcal{V}, \mathcal{E})$ be a graph, and $f : \mathcal{V} \to \mathbb{R}$ be a real-valued function. The inner product in the vertex space $\mathbb{R}^{\mathcal{V}}$ is defined as:*

$$\langle f, g \rangle_{\mathcal{V}} = \sum_{i=1}^n f(v_i) g(v_i), \tag{2}$$

*where $v_i$ denotes the $i$-th vertex in graph $\mathcal{G}$.*

In Hilbert space, the inner product introduces geometric notions such as "angle" and "length" between vectors. Similarly, the smoothness of node signals can be viewed as a geometric structure, where the Dirichlet energy function provides a means to quantify the smoothness of this geometric information.

According to Definition 2, if $f$ and $g$ are vector-valued functions, i.e., $f, g : \mathcal{V} \to \mathbb{R}^d$, the inner product can be extended as:

$$\langle f, g \rangle_{\mathcal{V}} = \sum_{i=1}^n \langle f(v_i), g(v_i) \rangle, \tag{3}$$

where $\langle \cdot, \cdot \rangle$ denotes the Euclidean inner product. In this Hilbert space 3, the inner product not only provides a geometric interpretation of the length and angles between vectors but also reflects the smoothness of node signals.

### 3.2 EDGE SPACE AS A GEOMETRIC PERSPECTIVE OF DIRICHLET ENERGY

The geometric structure of graph data is often embedded in the edge space. The topology of the edge space is determined by the adjacency matrix, while the geometric information is captured by the edge weights. The edge space can be viewed as a vector space where each edge corresponds to a basis vector. In this vector space, signals $X$ are represented by the differences across edges,

$\|f(v_i) - f(v_j)\|^2$. The Dirichlet energy essentially corresponds to the squared Euclidean norm of these difference vectors in the edge vector space.

Specifically, each term $\|f(v_i) - f(v_j)\|_2^2$ represents the signal variation across the edge $(v_i, v_j)$, and the total sum across all edges reflects the overall variation or energy of the signals on the entire graph. Hence, edge space provides a geometric lens to interpret Dirichlet energy through the differences across edges.

**Definition 3.2** (Linear Edge Space). *For a vector-valued function $f : \mathcal{V} \to \mathbb{R}^d$, the linear edge space $\mathcal{E}(f)$ is defined as:*

$$\mathcal{E}(f) = \sum_{(i,j) \in \mathcal{E}} a_{ij} \|f(v_i) - f(v_j)\|_2^2, \tag{4}$$

When $f$ is the identity mapping, i.e., $f(v_i) = x_i$, there exists a linear relationship between the linear edge space and the Dirichlet energy $E(X)$:

**Corollary 3.1** (Linear Relationship Between the Sum of Linear Edge Spaces and Dirichlet Energy).

$$\sum_{(i,j) \in \mathcal{E}} \mathcal{E}(f) = c \cdot E(X), \tag{5}$$

*where $c$ is a constant, and $\mathcal{E}$ is the corresponding Linear Edge Space 4.*

## 3.3 GEOMETRIC COLLAPSE INDUCED BY DIRICHLET LIMIT

Studies have shown that with each round of message passing in GNNs, the Dirichlet energy decays. Since Dirichlet energy is closely related to the edge space, it can be used to describe the geometric size of the edge space. When the Dirichlet energy approaches its limit, the geometric structure of the data collapses, meaning that the energy on some edges approaches zero, manifesting as over-smoothing.

In particular, Dirichlet energy plays a crucial role in training deep GNN models. As the number of layers increases, the Dirichlet energy continues to decay:

**Lemma 1.** The Dirichlet energy decays at a constant rate $c$:

$$E(X^{(l)}) \leq c^l \cdot E(X^{(l-1)}), \tag{6}$$

where $c \in [0, 1)$, indicating that the edge space of the graph shrinks progressively (proof is provided in the Appendix).

While small $E(X^{(l)})$ is associated with over-smoothing, excessively large values imply that node embeddings, even within the same class, are overly separated. For node classification tasks, each layer should maintain an appropriate level of Dirichlet energy to distinguish nodes across different classes while keeping nodes within the same class close. However, under certain conditions, theory proves that the upper bound of Dirichlet energy converges to zero as the number of layers tends to infinity Cai & Wang (2020), meaning all nodes collapse to a trivial fixed point in the embedding space, leading to the disappearance of the edge space.

Recent works Rusch et al. (2022b;a; 2023); Wu et al. (2023) define Dirichlet energy based on the random walk Laplacian matrix $\mathbf{\Delta}_{rw} = \mathbf{I}_n - \mathbf{D}^{-1}\mathbf{A}$ as $E_{rw}(\mathbf{X}) = \mathrm{tr}(\mathbf{X}^T \mathbf{\Delta}_{rw}\mathbf{X})$ and characterize over-smoothing as exponential convergence to a constant state, since the constant state corresponds to its null space. On the other hand, other research provides theoretical insights into convergence to the principal eigenvector, which is not always constant, as in GCN Kipf & Welling (2017).

We attribute these differences to the norm of $\mathbf{X}^{(k)}$, which obscures insights from Dirichlet energy. Similar to Dirichlet energy, norms are also constrained by the largest singular value of feature transformations:

**Proposition 3.2.** *(Graph Structure Irrelevance) Let $\mathbf{W} \in \mathbb{R}^{d \times d}$ be an arbitrary matrix with maximum singular value $\lambda_1^{\mathbf{W}}$, and $\phi$ be a component-wise non-expansive mapping satisfying $\phi(\mathbf{0}) = \mathbf{0}$. Then:*

$$|\phi(\mathbf{LXW})|_F \leq \lambda_1^{\mathbf{W}} \cdot |\mathbf{X}|_F, \tag{7}$$

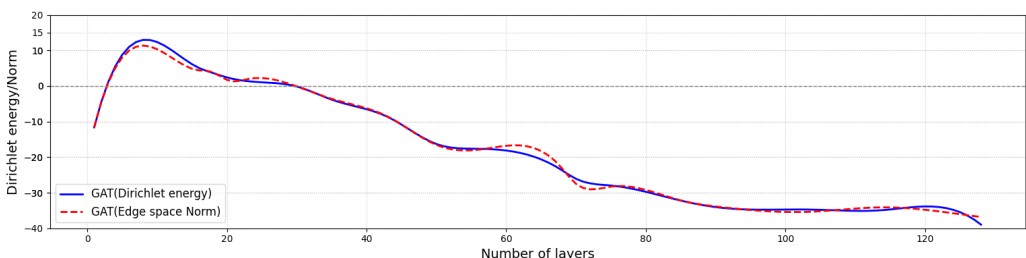

Figure 1: Analysis of the representations of the commonly used GAT model after $l$ layers on the Cora dataset, including Dirichlet energy and edge space length

Equation 9 indicates that when $\lambda_1^{\mathbf{W}^{(l)}} < 1$ for all layers, the feature maps converge to a zero matrix. Proof is provided in the Appendix.

The disappearance of Dirichlet energy is closely related to the collapse of the geometric structure of node embeddings. In Figure 1, we compare the Dirichlet energy with the total edge-space length and observe a strong correlation between them. This observation confirms the link between the disappearance of Dirichlet energy and geometric collapse. It provides an explanation for why some studies claim that GCNs converge to constant sequences Rusch et al. (2022b;a; 2023); Wu et al. (2023), or to values proportional to the degree of each node Oono & Suzuki (2019); Cai & Wang (2020); Zhou et al. (2021).

Since the norm of node embeddings obscures insights into Dirichlet energy, evaluating unnormalized energy alone is insufficient. Other metrics, such as MAD Chen et al. (2020a) and SMV Liu et al. (2020), have incorporated feature normalization to quantify over-smoothing. Furthermore, recent studies Di Giovanni et al. (2022); Maskey et al. (2023) have investigated Dirichlet energy in normalized settings as a means to better understand over-smoothing. Geometric collapse not only affects the norm of node embeddings but also severely impacts the relationships between nodes, which in turn affects the mutual information between them.

In GNNs, the mutual information between two nodes $v_i$ and $v_j$ can be expressed as $PMI(v_i, v_j) = f_\theta(\langle v_i, v_j \rangle)$, where $f_\theta$ is a function of the inner product. Given that node embeddings can be decomposed into magnitude and direction, $v = |v| \cdot \frac{v}{|v|}$, the inner product becomes $\langle v_i, v_j \rangle = |v_i| \cdot |v_j| \cdot \cos(\theta)$. This decomposition reveals the critical role of magnitude and direction in determining node correlations, explaining why pairwise distances based on embedding similarity are widely used to quantify over-smoothing Chen et al. (2020a); Zhao & Akoglu (2020b). Typically, nodes with smaller magnitudes are considered less important, further illustrating how geometric collapse, by shrinking both the magnitude and direction of embeddings, ultimately diminishes mutual information and the overall representational capacity of the network.

## 4 METHOD

In this section, we propose a novel GNN architecture, **CDE-GNN**, designed to effectively mitigate the over-smoothing problem prevalent in deep GNNs. Building on the theoretical analysis of Dirichlet energy and graph geometry provided earlier, CDE-GNN introduces an "initial Dirichlet energy" term to preserve the original topological information throughout the layers. This approach prevents excessive Dirichlet energy decay and maintains the discriminability of node embeddings. The initial Dirichlet energy is designed as a lower bound, ensuring that the geometric structure of the embeddings does not collapse during training.

### 4.1 NUMERICAL BEHAVIOR OF OVER-SMOOTHING AND TOPOLOGICAL COLLAPSE

As discussed earlier, over-smoothing is characterized by the continual decay of Dirichlet energy. As the number of GNN layers increases, the Dirichlet energy $E(X^{(l)})$ of node embeddings diminishes and may eventually approach zero. This trend leads to node embeddings becoming indistinguishable in high-dimensional space, exacerbating the over-smoothing problem. Furthermore, when the

Dirichlet energy $E(X^{(l)})$ approaches zero, the distances between node embeddings shrink, causing the entire network's topology to collapse, with all node embeddings converging to the same fixed point. This not only erases the discriminative information between nodes but also prevents deep GNNs from effectively capturing and leveraging the underlying geometric structure of the graph, severely limiting the model's expressiveness and performance. Specifically, topological collapse results in a lack of diversity in node embeddings, making it impossible to differentiate between nodes of different classes or structures, thereby negatively impacting downstream tasks.

## 4.2 INTRODUCING INITIAL DIRICHLET ENERGY AS A SOLUTION

To prevent Dirichlet energy from approaching zero during training and causing topological collapse, we propose incorporating the original graph topology as "initial Dirichlet energy." This energy is continuously injected into each layer. Specifically, CDE-GNN updates node embeddings in each layer by combining the feature aggregation of the current layer with the topological information from the initial layer to maintain the geometric structure of the embeddings. The initial Dirichlet energy serves as a lower bound for the Dirichlet energy, ensuring that even in deep networks, the geometric diversity of node embeddings is preserved, thereby preventing the embeddings from becoming overly homogeneous and the topology from collapsing.

### 4.2.1 LAYER-WISE UPDATE RULE IN CDE-GNN

Let $X^{(l)}$ denote node embeddings at layer $l$. The layer-wise update rule for CDE-GNN defined as:

$$X^{(l+1)} = \sigma\left(\tilde{\mathbf{L}}X^{(l)}\mathbf{W}^{(l)} + \alpha E_{\text{init}}X^{(l)}\right), \tag{8}$$

where $\tilde{\mathbf{L}}$ is the symmetrically normalized Laplacian matrix responsible for propagating and aggregating node features across the graph, $\mathbf{W}^{(l)}$ is the trainable weight matrix at layer $l$, and $\sigma(\cdot)$ is a non-linear activation function (e.g., ReLU). $E_{\text{init}}$ is the initial Dirichlet energy of the original graph.

The parameter $\alpha$ controls the contribution of the initial Dirichlet energy. By incorporating the term $\alpha E_{\text{init}}X^{(l)}$ at each layer, CDE-GNN ensures that the Dirichlet energy does not decay excessively, preserving the discriminability of node embeddings and preventing topological collapse. The initial Dirichlet energy $E_{\text{init}}$ captures the geometric information of the original graph and, when multiplied by the initial node embeddings $X^{(0)}$, ensures that each layer's update process retains the topological features of the original graph. This design enables the node embeddings to maintain sufficient geometric diversity even in deep networks, avoiding the tendency to collapse into a single fixed point. The initial Dirichlet energy acts as a lower bound during the training process, providing the necessary geometric constraints to ensure the model's stability and trainability in deep architectures.

## 5 EXPERIMENTS

In this section, we apply the CDE-GNN method to node classification tasks. Model hyperparameters are either adopted from publicly available literature or fine-tuned to improve classification accuracy. We use the Adam optimizer Kingma & Ba (2014) and employ an early-stopping strategy with a patience parameter of 200 epochs. Due to memory constraints on the OGBN-Arxiv dataset, we limit model depth to 32 layers. We also perform ablation studies to evaluate the performance of different configurations and empirically validate the theorems provided in Section 4. For all experiments, we use grid search to select hyperparameters. The primary loss function is cross-entropy, but for inductive learning on the PPI dataset, we use binary cross-entropy loss. Our implementation is based on PyTorch Paszke et al. (2019), PyTorch-Geometric Fey & Lenssen (2019), and Deep Graph Library (DGL) Wang (2019), and experiments are conducted on an Nvidia 3080 GPU.

In addition, we evaluate our model across various tasks and datasets (statistics provided in the Appendix), demonstrating that our model either outperforms or is competitive with other leading models in the field.

## 5.1 Node Classification

For this study, we use the Cora, Citeseer, and Pubmed datasets Sen et al. (2008), following the standard training/validation/test splits established by Yang et al. (2016), which include 20 nodes per class for training, 500 nodes for validation, and 1000 nodes for testing. Our training and evaluation procedures are consistent with Chen et al. (2020c), and we benchmark performance against a series of models, including GCN, GAT, Geom-GCN Pei et al. (2020), APPNP Klicpera et al. (2019), JKNet Xu et al. (2018b), WRGAT Suresh et al. (2021), PDE-GCN Eliasof et al. (2021), NSD Bodnar et al. (2022), GGCN Yan et al. (2022), H2GCN Zhu et al. (2020), DMP Yang et al. (2021), LINKX Lim et al. (2021), ACMII-GCN++ Luan et al. (2022), EGNN Zhou et al. (2021), and GCNII Chen et al. (2020c). The results summarized in Table 1 demonstrate the competitiveness and superiority of our model compared to existing methods.

Table 1: Node classification accuracy (%). Bold indicates the best performance, while underlining indicates the second-best performance. – indicates results were not available.

| Method | Cora | Citeseer | Pubmed | Squirrel | Film | Cham. | Corn. | Texas | Wisc. |
|---|---|---|---|---|---|---|---|---|---|
| GCN | 82.17 | 73.68 | 76.83 | 23.96 | 26.86 | 28.18 | 52.70 | 52.16 | 48.92 |
| GAT | 82.60 | 74.32 | 76.32 | 30.03 | 28.45 | 42.93 | 54.32 | 58.38 | 49.41 |
| GCNII | 82.72 | 77.20 | 79.00 | 38.47 | 32.87 | 60.61 | 74.86 | 69.46 | 74.12 |
| Geom-GCN | 79.50 | 77.99 | 78.75 | 38.32 | 31.63 | 61.57 | 60.81 | 67.57 | 64.12 |
| APPNP | 73.64 | 68.59 | 73.72 | 34.77 | – | 51.91 | 80.70 | 91.18 | – |
| JKNet | 79.48 | 75.85 | 77.64 | 44.72 | – | 62.92 | 66.73 | 75.53 | – |
| WRGAT | 82.47 | 76.81 | 77.22 | 48.85 | 36.53 | 65.24 | 81.62 | 83.62 | 86.98 |
| PDE-GCN | 82.83 | **78.75** | 78.63 | – | – | 66.01 | 89.73 | 93.50 | 91.95 |
| NSD | 81.37 | 78.00 | 78.19 | 56.34 | 37.79 | 68.68 | 86.49 | 85.95 | 89.41 |
| GGCN | 82.18 | 77.40 | 77.85 | 55.17 | 26.51 | **71.14** | 85.68 | 84.86 | 86.86 |
| H2GCN | 82.10 | 77.13 | 78.19 | 36.48 | 35.70 | 60.11 | 82.70 | 84.86 | 87.65 |
| DMP | 80.75 | 76.87 | 77.97 | 47.26 | 35.72 | 62.28 | 89.19 | 89.19 | 80.86 |
| LINKX | 78.87 | 73.19 | 76.56 | **61.81** | 36.10 | 68.42 | 77.84 | 74.60 | 75.49 |
| ACMII-GCN++ | 82.72 | 77.12 | 78.41 | – | 37.09 | – | 86.49 | 88.38 | 88.43 |
| CDE-GNN | **83.54** | 78.13 | **79.52** | 59.41 | **39.50** | 70.02 | **91.35** | **94.80** | **92.35** |

The results in Table 1 show that our model achieves either the best or second-best classification accuracy on Cora, Citeseer, Pubmed, and several other datasets, demonstrating its strong competitiveness. For instance, on the Cora and Pubmed datasets, CDE-GNN achieves accuracy of **83.54%** and **79.52%**, respectively, outperforming other baseline models. Additionally, CDE-GNN performs well on heterogeneous graph datasets like Squirrel, Film, and Chameleon, illustrating its adaptability across diverse graph structures.

We also analyze model accuracy across different numbers of layers (ranging from 2 to 64), as shown in Table 2. The analysis reveals that CDE-GNN is resilient to over-smoothing, even with an increasing number of layers.

Beyond semi-supervised settings, we evaluate our model in fully supervised node classification tasks, including both homophilic and heterophilic datasets, as categorized in Pei et al. (2020). We apply our model to datasets including Cora, Citeseer, Pubmed, Chameleon Rozemberczki et al. (2021), Film, Cornell, Texas, and Wisconsin, following consistent splits of 48%, 32%, and 20% for training, validation, and testing, respectively. As per Pei et al. (2020), we report average performance over 10 random splits and compare against models such as GCN, GAT, Geom-GCN, APPNP, JKNet, Inception, GCNII, and PDE-GCN. The results are detailed in Table 1, where we observe improvements in accuracy over other considered methods.

**Comparison with State-of-the-Art Models.** To validate the effectiveness of our proposed method, we conduct a series of semi-supervised node classification experiments, comparing our model with several competitive deep GCN models, as shown in Table 2. Notably, our proposed model improves upon the previous state-of-the-art by an average of 1%. However, earlier deep GCN architectures (e.g., JKNet) did not significantly outperform shallow models. By contrast, GCNII improved performance by 2% over previous methods, clearly demonstrating the effectiveness and

advantages of deep GCNs. In this paper, we further enhance deep GCN performance by introducing optimal residual connections in each layer, highlighting the benefits of deep network structures.

Table 2: Node classification accuracy (%) for different depths: 2, 16, and 32/64 layers. The best accuracy in each column is highlighted in bold.

| Dataset | Cora | | | Pubmed | | | Coauthor-Physics | | | OGBN-Arxiv | | |
|---|---|---|---|---|---|---|---|---|---|---|---|---|
| Layers | 2 | 16 | 64 | 2 | 16 | 64 | 2 | 16 | 32 | 2 | 16 | 32 |
| GCN | 82.5 | 22.0 | 21.9 | **79.7** | 37.9 | 38.4 | 92.4 | 13.5 | 13.1 | 70.4 | 70.6 | 68.5 |
| SGC | 75.7 | 72.1 | 24.1 | 76.1 | 70.2 | 38.2 | 92.2 | 91.7 | 84.8 | 69.2 | 64.0 | 59.5 |
| JKNet | 80.8 | 74.5 | 70.0 | 77.2 | 70.0 | 66.1 | 92.7 | 92.2 | 91.6 | **70.6** | 71.8 | 71.4 |
| APPNP | 82.9 | 79.4 | 79.5 | 79.3 | 77.1 | 76.8 | 92.3 | 92.7 | 92.6 | 68.3 | 65.5 | 60.7 |
| GCNII | 82.4 | 84.6 | 85.4 | 77.5 | 79.8 | 79.9 | 92.5 | 92.9 | 92.9 | 70.1 | 71.5 | 70.5 |
| EGNN | 83.2 | 85.4 | 85.7 | 79.2 | 80.0 | 80.1 | 92.6 | 93.1 | 93.3 | 68.4 | 72.7 | 72.7 |
| CDE-GNN | **83.5** | **86.4** | **86.6** | 79.5 | **80.2** | **80.8** | **92.9** | **93.5** | **94.2** | 68.9 | **72.8** | **72.8** |

**Detailed Comparison with Other Deep Models.** As shown in Table 2, the results across different depths of deep models can be summarized as follows: Our model, CDE-GNN, consistently outperforms all baseline models on every dataset, with significant performance improvements as the model depth increases. Specifically, CDE-GNN achieves classification accuracy of **86.6%**, **80.8%**, and **94.2%** on the Cora, Pubmed, and Coauthor-Physics datasets, respectively, at 64 layers, indicating that deep GNN architectures can effectively leverage optimal residuals. In contrast, other state-of-the-art deep models, such as SGC, JKNet, and APPNP, often suffer from performance degradation as the number of layers increases, sometimes even performing worse than shallow models. This demonstrates that traditional deep GNN architectures are still significantly affected by the over-smoothing problem.

As one of the most competitive deep architectures in the literature, GCNII enhances the preservation of identity mappings by amplifying the smallest singular value of the weight matrix. Meanwhile, EGNN introduces orthogonal weight initialization and applies orthogonal weight regularization based on an upper bound of Dirichlet energy to balance identity mappings with task adaptation. By combining these two methods, CDE-GNN introduces optimal residuals at each layer, further boosting model performance. Notably, even at a depth of 64 layers, CDE-GNN continues to exhibit performance improvements.

CDE-GNN surpasses GCNII and EGNN on small-scale datasets like Cora, Pubmed, and Coauthor-Physics and demonstrates significant advantages on large-scale datasets such as OGBN-Arxiv. Specifically, CDE-GNN achieves an accuracy of **72.8%** on OGBN-Arxiv, substantially outperforming other baseline models. These experimental results strongly validate that by introducing optimal residuals and controlling the convergence of Dirichlet energy, we can effectively address the over-smoothing problem and extend traditional GCNs into deep architectures.

## 5.2 ABLATION STUDY

**Hyperparameter Analysis.** We conduct ablation studies to explore the impact of different hyperparameters on model performance, specifically focusing on activation functions, dropout rates, and the number of hidden units. The results, shown in Table 3, are evaluated for significance.

As seen in Table 3, the size of the hidden layers has a noticeable effect on model performance. Typically, larger hidden layers (e.g., 64 units) yield slightly better performance across most datasets, while smaller hidden layers (e.g., 16 units) also achieve comparable performance on some datasets. Specifically, the larger hidden layer size (64 units) generally produces higher accuracy on datasets such as Cora, Pubmed, and OGBN-Arxiv. However, smaller hidden layers (16 units) achieve strong performance on the Physics dataset. Therefore, selecting an appropriate hidden layer size should balance the trade-off between performance and computational cost, depending on the dataset's characteristics. For datasets with a high number of classes, such as OGBN-Arxiv, a larger number of hidden units is recommended.

Table 3: Ablation study results across activation functions, dropout rates, and hidden unit sizes.

| Component | Type | Cora | | | Pubmed | | | Physics | | | OGBN-Arxiv | | |
|---|---|---|---|---|---|---|---|---|---|---|---|---|---|
| | | 2 | 16 | 64 | 2 | 16 | 64 | 2 | 16 | 32 | 2 | 16 | 32 |
| Activation | ReLU | 83.5 | 86.4 | 86.6 | 79.5 | 80.2 | 80.8 | 92.9 | 93.5 | 94.4 | 68.9 | 71.9 | 72.8 |
| | Sigmoid | 53.5 | 56.0 | 56.3 | 49.5 | 49.7 | 49.3 | 62.9 | 62.5 | 63.0 | 14.6 | 13.2 | 11.8 |
| | None | 76.7 | 75.1 | 74.7 | 76.4 | 75.5 | 73.1 | 84.8 | 86.4 | 86.3 | 63.2 | 63.7 | 64.2 |
| Dropout | 0.2 | 82.7 | 85.9 | 85.8 | 78.9 | 80.1 | 80.8 | 93.3 | 93.9 | 94.4 | 68.8 | 70.8 | 71.6 |
| | 0.4 | 83.1 | 86.2 | 86.6 | 79.0 | 80.1 | 80.5 | 92.7 | 93.6 | 93.2 | 66.6 | 65.7 | 65.0 |
| | 0.6 | 80.0 | 78.8 | 74.1 | 79.6 | 79.4 | 78.9 | 90.5 | 87.9 | 67.5 | 65.1 | 64.3 | 64.0 |
| Hidden | 16 | 83.5 | 86.1 | 86.3 | 79.7 | 80.2 | 80.5 | 92.8 | 93.6 | 94.4 | 68.9 | 71.6 | 72.4 |
| | 32 | 84.0 | 86.2 | 86.3 | 80.0 | 80.3 | 80.8 | 93.1 | 93.5 | 94.3 | 69.7 | 72.0 | 72.8 |
| | 64 | 84.2 | 86.4 | 86.6 | 80.2 | 80.3 | 80.7 | 93.3 | 93.8 | 94.1 | 69.6 | 71.9 | 72.7 |

For activation functions, the non-linear ReLU consistently achieves the best results across all experiments. Compared to the Sigmoid function, ReLU better handles non-linear relationships and mitigates the vanishing gradient problem in the saturated regions of Sigmoid. Additionally, ReLU is computationally more efficient, as Sigmoid involves expensive exponential operations. Without an activation function, the model is limited to learning only linear relationships, restricting its ability to adapt to complex non-linear data.

Regarding dropout rates, we observe that varying dropout rates have a significant impact on model performance. Typically, a lower dropout rate (e.g., 0.2) yields better performance, while higher dropout rates (e.g., 0.6) lead to performance degradation. Specifically, the lower dropout rate (0.2) performs best across most datasets and model components, particularly on Cora and Pubmed. In practice, this parameter should be adjusted and validated according to the specific dataset. Moderate dropout helps reduce overfitting and enhances the model's generalization ability, but too high a dropout rate may result in information loss and degraded performance.

**Impact of Activation Functions.** As shown in Table 3, using ReLU as the activation function significantly improves model performance across all datasets, particularly on Cora and Pubmed, where ReLU outperforms Sigmoid and the absence of an activation function. This underscores the importance of activation functions in enhancing the non-linear expressiveness of the model.

**Optimizing Dropout Rates.** By comparing different dropout rates, we find that a 0.2 dropout rate strikes a balance between preventing overfitting and preserving sufficient information flow, enabling the model to learn effectively. However, higher dropout rates (e.g., 0.6) excessively reduce information flow, impairing model performance.

**Choosing the Number of Hidden Units.** Increasing the number of hidden units generally leads to improved performance, but this improvement varies across datasets. For instance, on Cora and Pubmed, increasing the number of hidden units to 64 improves performance, while on Physics, a hidden unit size of 16 achieves competitive results. Thus, choosing the appropriate number of hidden units should depend on the nature of the dataset.

In summary, the ablation study demonstrates that appropriate choices of activation functions, dropout rates, and hidden unit sizes significantly influence the performance of CDE-GNN. Optimizing these hyperparameters is crucial for maximizing the model's overall performance.

## 6 CONCLUSION

This paper presents a geometric perspective on the over-smoothing and compression issues in deep GNNs, revealing how increasing depth leads to indistinguishable node embeddings and adversely affects model performance. We propose a geometric framework based on a parameterized graph Laplacian operator, which controls the lower bound of Dirichlet energy to prevent geometric collapse and mitigate over-smoothing. Both theoretical analysis and empirical results demonstrate that this method significantly enhances the trainability and performance of deep GNNs, particularly in node classification tasks, outperforming existing state-of-the-art methods. Future work could explore extending the application of geometric information to heterogeneous and dynamic graphs.

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
