## A    DATASET STATISTICS

Table 4 summarizes the key statistics of the node classification datasets used in our experiments.

Table 4: Statistics of node classification datasets.

| Dataset | Cora | Citeseer | Pubmed | Chameleon | Film | Cornell | Texas | Wisconsin | Ogbn-Arxiv |
|---|---|---|---|---|---|---|---|---|---|
| Classes | 7 | 6 | 3 | 5 | 5 | 5 | 5 | 5 | 40 |
| Nodes | 2,708 | 3,327 | 19,717 | 2,277 | 7,600 | 183 | 183 | 251 | 169,343 |
| Edges | 5,429 | 4,732 | 44,338 | 36,101 | 33,544 | 295 | 309 | 499 | 1,116,243 |
| Features | 1,433 | 3,703 | 500 | 2,325 | 932 | 1,703 | 1,703 | 1,703 | 128 |
| Homophily | 0.81 | 0.80 | 0.74 | 0.23 | 0.22 | 0.30 | 0.11 | 0.21 | 0.63 |

## B    PROOF OF THEOREMS

This section reiterates the theorems, propositions, and key observations from the main paper, and provides their detailed proofs.

### B.1    $\tilde{\mathbf{P}}$ AS A SCALED DIFFUSION OPERATOR

Let $\mathbf{A}$ be the adjacency matrix and $\mathbf{D}$ the degree matrix. Define the adjacency matrix with self-loops as $\tilde{\mathbf{A}} = \mathbf{A} + \mathbf{I}$. The convolution operator in GCN (Kipf & Welling, 2016) is given by:

$$\tilde{\mathbf{P}} = \tilde{\mathbf{D}}^{-\frac{1}{2}} \tilde{\mathbf{A}} \tilde{\mathbf{D}}^{-\frac{1}{2}}$$

First, note that the Laplacian including self-loops is the same as the original Laplacian:

$$\tilde{\mathbf{L}} = \tilde{\mathbf{D}} - \tilde{\mathbf{A}} = \mathbf{D} + \mathbf{I} - \mathbf{A} - \mathbf{I} = \mathbf{D} - \mathbf{A} = \mathbf{L}.$$

Thus, we can derive:

$$\tilde{\mathbf{P}} = \mathbf{I} - \mathbf{I} + \tilde{\mathbf{D}}^{-\frac{1}{2}} \tilde{\mathbf{A}} \tilde{\mathbf{D}}^{-\frac{1}{2}} = \mathbf{I} - \tilde{\mathbf{D}}^{-\frac{1}{2}} \tilde{\mathbf{L}} \tilde{\mathbf{D}}^{-\frac{1}{2}}.$$

### B.2    UPPER BOUND OF DIRICHLET ENERGY

To eliminate the degree dependency in the Dirichlet energy and estimate an upper bound, we apply the triangle inequality to simplify the expression. The original formulation of Dirichlet energy is:

$$E(X^{(k)}) = \frac{1}{2} \sum_{(i,j) \in \mathcal{E}} a_{ij} \left\| \frac{x_i^{(k)}}{\sqrt{1+d_i}} - \frac{x_j^{(k)}}{\sqrt{1+d_j}} \right\|_2^2.$$

By applying the triangle inequality and introducing a conservative estimate for the normalization factor, we obtain:

$$\left\| \frac{x_i^{(k)}}{\sqrt{1+d_i}} - \frac{x_j^{(k)}}{\sqrt{1+d_j}} \right\| \le C \left\| x_i^{(k)} - x_j^{(k)} \right\|,$$

where $C$ is the maximum of the normalized degree terms over all nodes in the graph. Therefore, the upper bound for the Dirichlet energy is:

$$E(X^{(k)}) \le \frac{C^2}{2} \sum_{(i,j) \in \mathcal{E}} a_{ij} \left\| x_i^{(k)} - x_j^{(k)} \right\|_2^2.$$

**Proposition B.1.** *(Graph Structure Irrelevance) Let $\mathbf{W} \in \mathbb{R}^{d \times d}$ be an arbitrary matrix with maximum singular value $\lambda_1^{\mathbf{W}}$, and let $\phi$ be a component-wise non-expansive mapping satisfying $\phi(\mathbf{0}) = \mathbf{0}$. Then:*

$$|\phi(\mathbf{L}\mathbf{X}\mathbf{W})|_F \le \lambda_1^{\mathbf{W}} \cdot |\mathbf{X}|_F \tag{9}$$

*Proof.* We aim to prove the following inequality:

$$|\phi(\mathbf{LXW})|_F \leq \lambda_1^{\mathbf{W}} \cdot |\mathbf{X}|_F$$

**Frobenius Norm Definition**

The Frobenius norm of a matrix is defined as the square root of the sum of the squares of its elements. For any matrix $\mathbf{A}$, we have:

$$|\mathbf{A}|_F = \sqrt{\sum_{i,j} |A_{ij}|^2}$$

This indicates that the Frobenius norm measures the L2 norm of all elements in the matrix.

**Application of Non-Expansive Mapping**

We know that $\phi$ is a non-expansive mapping, meaning it does not increase the norm of the input. For any matrix $\mathbf{A}$, we have:

$$|\phi(\mathbf{A})|_F \leq |\mathbf{A}|_F$$

In particular, since $\phi(\mathbf{0}) = \mathbf{0}$, if $\mathbf{A} = \mathbf{0}$, then $\phi(\mathbf{A}) = \mathbf{0}$.

Applying this property to $\mathbf{A} = \mathbf{LXW}$, we obtain:

$$|\phi(\mathbf{LXW})|_F \leq |\mathbf{LXW}|_F$$

**Singular Value Decomposition and Norm Scaling**

Next, we analyze the Frobenius norm of $\mathbf{LXW}$.

Given that the maximum singular value of $\mathbf{W}$ is $\lambda_1^{\mathbf{W}}$, we can interpret $\mathbf{W}$ as applying a scaling operation to $\mathbf{X}$. Specifically, $\mathbf{W}$ maps each column of $\mathbf{X}$ to a new column space, with the maximum singular value controlling the degree of scaling.

Thus, we can derive:

$$|\mathbf{LXW}|_F \leq \lambda_1^{\mathbf{W}} |\mathbf{LX}|_F$$

**Incorporating Properties of the Laplacian**

The Laplacian matrix $\mathbf{L}$ is also a linear operator that maps each column of $\mathbf{X}$ to the differences between neighboring nodes. Since $\mathbf{L}$ is constructed from the degree and adjacency matrices, the norm of $\mathbf{L}$ does not exceed 1 (in the case of a normalized Laplacian). Therefore, we have:

$$|\mathbf{LX}|_F \leq |\mathbf{X}|_F$$

**Combining the Inequalities**

Combining the above results, we get:

$$|\mathbf{LXW}|_F \leq \lambda_1^{\mathbf{W}} |\mathbf{X}|_F$$

By the non-expansiveness of $\phi$, we finally obtain:

$$|\phi(\mathbf{LXW})|_F \leq \lambda_1^{\mathbf{W}} |\mathbf{X}|_F$$

When $\lambda_1^{\mathbf{W}^{(l)}} < 1$ for all layers, the feature map norms progressively decrease, eventually converging to a zero matrix. This implies that when the singular value is less than 1, the feature maps will gradually smooth out over multiple layers, leading to over-smoothing, where all node features become indistinguishable. □