# OpenReview forum: "CHARGE DIRICHLET ENERGY: Geometric Perspectives on Over-smoothing in Deep Graph Neural Networks"
_ICLR.cc/2025/Conference — Submitted to ICLR 2025_

### Official Review · Reviewer_JWcE · 2024-10-16

**Soundness:** 2
**Presentation:** 2
**Contribution:** 1
**Rating:** 3
**Confidence:** 5

**Summary:**

This paper proposes a new understanding of Dirichlet energy in the context of GNNs, revealing the relationship between Dirichlet energy decay and edge space collapse.

The paper also introduces a new message updating scheme, which prevents over-smoothing by incorporating a residual term weighted by the minimum Dirichlet energy. And the authors conducted extensive comparison experiments demonstrating the effectiveness.

**Strengths:**

The authors revealed that the Dirichlet energy decay in deep GNNs is linearly proportional to the edge space sum with a constant $c$.

The authors provided that, within the Dirichlet energy analystic framework, it is crucial for the activation function $\phi$ to satisfy $\phi(0)=0$ in Proposition 3.2. This may indirectly explains why adding a shift $b$ in SReLU is effective.

The minimum Dirichlet energy constrained message updating scheme showed a good performance.
Extensive benchmark performance comparisons.

**Weaknesses:**

**Lack of novelty**

This paper is a direct follow-up of [1].

Consider equation(8) in [1] :

$X^{(k)}=\sigma\left(\left[\left(1-c_{\min }\right) \tilde{P} X^{(k-1)}+\alpha X^{(k-1)}+\beta X^{(0)}\right] W^{(k)}\right)$,

where $\alpha+\beta=c_{\text{min}}$.

When set $\beta=0$ and rephrase the equation as

$X^{(k)}=\sigma\left(\left[ \tilde{P} X^{(k-1)}+ \frac{c_{\text{min}}}{\left(1-c_{\min }\right)} X^{(k-1)} \right]W^{(k)}\right)$.

Replacement the symbol $\frac{1}{1-c_{\text{min}}} \to \alpha$  and $c_{\text{min}} \to E_{init}$ , we immediately obtain:

$X^{(k+1)}=\sigma( \tilde{P} X^{(k)}W^{(k)} + \alpha E_{init} X^{(k)}W^{(k)}))$.

Compare to Equation(8) in this paper:

$X^{(l+1)}=\sigma\left(\tilde{\mathbf{L}} X^{(l)} \mathbf{W}^{(l)}+\alpha E_{\text {init }} X^{(l)}\right)$

These two update functions are remarkably similar, except for a weight matrix.

It has been demonstrated in [1] that, utilizing two distinct forms of residual connections can effectively constrain the lower bound of the DIRICHLET energy, and the constraint's intensity can be modulated by a gating parameter.

The approach presented in this paper appears to gracefully fit within this previously established framework, representing a specific instance when $\beta=0$.

The main contribution is addressing the issue that researchers do not know how to choose an appropriate lower bound $c_{\text{min}}$ for different datasets. While in this paper, the authors suggest that simply using the initial DIRICHLET energy $E_{\text{init}}$ as the lower bound works very well.

The authors need to clarify how their method distinguishes itself from or enhances the previous approach.
It would be more valuable if the authors could elaborate on the significance of omitting the weight matrix in the update function and the rationale behind selecting the initial DIRICHLET energy as the lower bound among various initial value choices.

[1] K. Zhou *et al.*, “Dirichlet energy constrained learning for deep graph neural networks,” in *Advances in Neural Information Processing Systems*, M. Ranzato, A. Beygelzimer, Y. Dauphin, P. S. Liang, and J. W. Vaughan, Eds., Curran Associates, Inc., 2021, pp. 21834–21846.



**Lack of experiment**

- Dirichelet energy visualization.  Plotting the Dirichlet energy of each layer with and without the $E_{init}$ term would be more persuasive.  As stated in Section 4.2, 'The initial Dirichlet energy serves as a lower bound for the Dirichlet energy,' it is expected that  $E_{\text{Dirichlet}}$ will converge to the lower bound $E_{init}$ as the number of layers increases.

- The claim that  $E_{init}$ preventing topological collapse in Section 4.2 should be supported by experimental evidence. Visualizing the node representations in the final layer and comparing them to the initial topology would be helpful. Consider using commonly employed techniques, such as t-SNE visualization\[2\]\[3\] or a color-propagation test[4].




\[2\]D. Shen, C. Qin, Q. Zhang, H. Zhu, and H. Xiong, “Handling over-smoothing and over-squashing in graph convolution with maximization operation,” *IEEE Trans. Neural Netw. Learn. Syst.*, pp. 1–14, 2024, doi: [10.1109/TNNLS.2024.3442270](https://doi.org/10.1109/TNNLS.2024.3442270).

\[3\]M. Liu, H. Gao, and S. Ji, “Towards Deeper Graph Neural Networks,” in *KDD*, Aug. 2020, pp. 338–348. doi: [10.1145/3394486.3403076](https://doi.org/10.1145/3394486.3403076).

\[4\]K. Xu, C. Li, Y. Tian, T. Sonobe, K. Kawarabayashi, and S. Jegelka, “Representation learning on graphs with jumping knowledge networks,” in *International conference on machine learning*, PMLR, 2018, pp. 5453–5462.

**Questions:**

**General Concerns**

- Is $\alpha$ in Equation(8) a learnable parameter or a manually defined hyper-parameter? The authors did not clarify this.
- What are the exact values of  $\alpha$ and   $E_{init}$ for each dataset in the experiments? These details are not mentioned in either the main text or the supplementary materials.

- If  $\alpha$  is a hyper-parameter,  the model's performance under different $\alpha$ settings should be reported. Additionally, a discussion on how to select an appropriate $\alpha$ would provide valuable insights for the community.
- Why not include a comparison with EGNN in Table 1?
- Why not use SReLU as the activation function in Section 5.2, given that it has been demonstrated in EGNN[1] for preserving Dirichlet energy, especially on large datasets like OGBN-arxiv?
- It would be helpful if the authors could clarify the source of the baseline model performances in Table 1. Specifically, did they conduct all the experiments themselves, or were some of the results sourced from other papers?
- Given that the benchmark results are based on 10 random splits, would it be possible to provide the standard deviation in addition to the mean? This could offer a more comprehensive understanding of the results.
- It would be better to include publicly available code to ensure reproducibility;

\[1\] K. Zhou et al., “Dirichlet energy constrained learning for deep graph neural networks,” in Advances in Neural Information Processing Systems, M. Ranzato, A. Beygelzimer, Y. Dauphin, P. S. Liang, and J. W. Vaughan, Eds., Curran Associates, Inc., 2021, pp. 21834–21846.

\[2\] J. Zhu, Y. Yan, L. Zhao, M. Heimann, L. Akoglu, and D. Koutra, “Beyond Homophily in Graph Neural Networks: Current Limitations and Effective Designs,” in *Advances in Neural Information Processing Systems*, H. Larochelle, M. Ranzato, R. Hadsell, M. F. Balcan, and H. Lin, Eds., Curran Associates, Inc., 2020, pp. 7793–7804.


**Ambiguous statement**

- In Eqaution(8),  $E_{init}$ is **multiplied by $X^{(l)}$** .  While in Line 304, it states **multiplied by $X^{(0)}$** .

  > Line 304
  > The initial Dirichlet energy Einit captures the geometric information of the original graph and, when multiplied by the initial node embeddings X(0), ensures that each layer’s update process retains the topological features of the original graph.

- data split question. What are the actual splits used for the Cora, Citeseer, and PubMed datasets? If Yang's split was adopted, why not use the same split as Geom-GCN for consistency?
  > Line 326
  >
  > For this study, we use the **Cora, Citeseer, and Pubmed** datasets Sen et al. (2008), **following the standard training/validation/test splits established by Yang et al. (2016)**
  >
  > Line 367
  >
  > We apply our model to datasets including **Cora, Citeseer, Pubmed**, Chameleon Rozemberczki et al.(2021), Film, Cornell, Texas, and Wisconsin, **following consistent splits of 48%, 32%, and 20%** for training, validation, and testing, respectively.


**Minor comments**

- Table1, Table 2, Table 3 out of page width;
- In the introduction  section, authors use the notion `a minimum Dirichlet energy ω` . But in the following text, $\omega$  is no longer used; instead, $E_{init}$ is used.  A consistent notation across the whole paper would be better;
- There is a mistake in Equation (5):  $\mathcal{E}(f)$ is already a summation over the  (i,j) pairs and cannot be summed again.

---

### Official Review · Reviewer_z88i · 2024-10-28

**Soundness:** 2
**Presentation:** 1
**Contribution:** 3
**Rating:** 3
**Confidence:** 3

**Summary:**

Inspired by previous work on over-smoothing and Dirichlet energy, the authors propose a simple, intuitive, and generally applicable method named CDE-GNN to address the over-smoothing problem in Graph Neural Networks. The proposed approach is validated by theoretical and empirical results.

**Strengths:**

1. The paper proposed a very intuitive and generally applicable idea to effectively alleviate the over-smoothing problem for GNNs.
2. The proposed idea is well-motivated by solid theoretical insights and results.
3. The paper is easy to read.

**Weaknesses:**

1. Some theoretical terms in Section 3 are not sufficiently introduced and explained. The authors are suggested to provide more detailed background and solid definition of the quantities involved.

2. Several parts are repetitive or even inconsistent. For example, Section 4.1 is redundant and repetitive of the earlier content, as well as the Hyperparameter Analysis with its three following paragraphs in Section 5.2. Please refer to the Questions for details.

3. The hyperparameter analysis is not insightful. The study of different activation functions, hidden dimensions, and dropout rates is old-fashioned and not unique to the proposed approach.

**Questions:**

1. In Equation (5), why summing over the edges twice?

2. In Line 229-230, should Equation 9 actually refer to Equation 7?

3. In Line 080, the authors state that the energy lower bound is learnable. However, as in Section 4.2.1, it is fixed as the initial energy. How is it learnable?

4. In Line 304, it says the initial energy is multiplied by the initial embeddings, whereas in Equation 8 it is multiplied by the embeddings per layer. Which one is correct?

5. In Table 1, why are there two bolded results for Citeseer and Film? Also, are those results for the semi-supervised or fully-supervised setting? It says in Line 334 and 370 that both settings’ results are in Table 1 but there are no marks for different settings.

6. Several results in Table 2 and 3 don’t match. For example, on the Physics dataset with 32 layers, the optimal accuracy is 94.2 and 94.4, respectively. Why?

---

### Official Review · Reviewer_iAkJ · 2024-11-05

**Soundness:** 2
**Presentation:** 2
**Contribution:** 1
**Rating:** 3
**Confidence:** 5

**Summary:**

This paper discusses an approach to alleviate over-smoothing of deep graph neural networks through the lens of Dirichlet energy. The idea lies in adding one additional term in the layer-wise propagation which takes into account the Dirichlet energy of the initial graph. Experiments have been conducted on several node classification benchmarks showing that the model can yield better performance with increasing depth of the graph neural network.

**Strengths:**

1. The problem tackled is important and this paper approaches it through the concept of Dirichlet energy.

2. The method seems to offer empirical enhancements on various benchmarks.

**Weaknesses:**

1. Though a lot of efforts have been paid on discussing DIrichlet energy, how the proposed approach is linked to preserving Dirichlet energy still remains very unclear. Eq. (8) was introduced alone while more theoretical investigations and experimental observations should be incorporated.

2. The presentation needs improvement. Wordy sentences present at times with many of them constantly repeated, e.g., the contents in Sec. 4.1 has been discussed multiple times in the previous sections and should be simplified for more informative contents.

3. Some concepts were introduced with confusion and did not exhibit strong correlation to the proposed approach. For instance, how Proposition 3.2 is related to Eq. (8) (e.g., how Eq. (8) help address the vanishing problem) is unclear. There is also no clear reason of introducing the edge space (Definition 3.2) and Corollary 3.1 conveys limited information.

4. Experiment settings are not convincing. For example, the reported performance of the baselines are remarkably lower than the official leaderboard of obgn-arxiv (https://ogb.stanford.edu/docs/leader_nodeprop/#ogbn-arxiv).

Minor:

Misuse of citet vs citep in multiple places hinders the readability. The authors are encouraged to correct these presentation issues.

Overall, the paper in its current shape is unsatisfactory in justifying the rationale of the proposed approach, which is simply Eq. (8), and relevant discussions in both theory and experiment are missing, making it less self-consistent.

**Questions:**

1. How Eq. (8) could help preserve Dirichlet energy of each layer? By simply multiplying $X^{(l)}$ with a scalar and adding that to before applying nonlinearity, how would it guarantee Dirichlet energy is not vanishing?

2. How is the approach compared with others like GCNII in terms of preserving DIrichlet energy? Intuitively adding initial residuals can already effectively preserve Dirichlet energy. Why the proposed approach adds the additional term before the nonlinearity and why the embedding of the previous layer instead of the initial layer is used?

3. Why is the performance of all models on ogbn-arxiv much lower than officially reported results?

4. Are there any plots of the layer-wise Dirichlet energy of the proposed model as well as some baselines (e.g., GCNII [1]) on these benchmarks? How does Dirichlet energy connect to actual performance? This would be important to help readers gain more intuition and also help understand the efficacy of the proposed approach.

5. What is the rationale of introducing edge space (Definition 3.2) and how does it play a role in justifying the proposed method?


[1] Chen et al. Simple and Deep Graph Convolutional Networks. ICML'20.

---

### Official Review · Reviewer_Wyan · 2024-11-07

**Soundness:** 2
**Presentation:** 2
**Contribution:** 1
**Rating:** 3
**Confidence:** 4

**Summary:**

This paper studies the over-smoothing problem of deep graph neural networks and proposes a geometric perspective for addressing over-smoothing. Specifically, the authors analyze the Dirichlet energy minimized by the feed-forward computation process of GCNs and propose a new method based on Dirichlet energy for resolving over-smoothing when the layer number increases. Experiments on small datasets demonstrate the effectiveness of the model.

**Strengths:**

1. The paper is well motivated and studies an important problem from an interesting perspective

2. The paper is generally well written and easy to follow

**Weaknesses:**

1. The proposed method has limited novelty given existing works that have explored similar ideas and model designs, e.g., [1,2]. Adding self-loop or residual link or strengthening the information of the centered nodes have been extensively used by existing GNN models, such as the well-known ones [1, 2]

2. The theoretical results are not new and have been derived in the literature, e.g., [3, 4]. The result of Lemma 1 has been proved in [3] and [4]. Besides, the analysis presented in this paper only shows the result that is already well-known, i.e., over-smoothing will happen when the layer increases. There lacks analysis in why and how the propose model can address the over-smoothing.

3. The experimental evaluation is limited in small datasets and comparison with state-of-the-arts is insufficient. More experiments on large datasets such as ogbn-products and ogbn-proteins are suggested. More comparison with state-of-the-art GNNs, especially the ones that can overcome over-smoothing, e.g., GCNII, are needed.

[1] Simple and Deep Graph Convolutional Networks, ICML 2020

[2] Predict then Propagate: Graph Neural Networks meet Personalized PageRank, ICLR 2019

[4] A note on over-smoothing for graph neural networks. Arxiv 2020

[5] Dirichlet Energy Constrained Learning for Deep Graph Neural Networks, NeurIPS 2022

**Questions:**

1. How does the model perform on large graph datasets, such as ogbn-products and ogbn-proteins?

2. Can the authors provide validation on whether the over-smoothing problem is indeed addressed in practice on the experimental datasets?

---

### Official Review · Reviewer_DPwo · 2024-11-10

**Soundness:** 3
**Presentation:** 2
**Contribution:** 2
**Rating:** 5
**Confidence:** 4

**Summary:**

The paper addresses the problem of over-smoothing in deep GNNs, where node embeddings become indistinguishable as network depth increases, leading to degraded performance. The authors present a novel geometric perspective on this issue and propose a method called Charge Dirichlet Energy (CDE-GNN). The authors validate their method through comprehensive experiments across various datasets and network depths, showing consistent performance improvements over baseline models.

**Strengths:**

1. Well-structured presentation progressing from problem motivation to theoretical analysis to practical solution.
2. Comprehensive empirical validation across multiple datasets.

**Weaknesses:**

1. Limited Novelty:  using Dirichlet energy to overcome oversmoothness has been extensively studied.
2. lack of detailed analysis of computational overhead compared to baseline methods

**Questions:**

1. The layer propagation rule shows strong similarity to EGNN's Lower-bounded Residual Connection [1]. The paper needs to better elaborate on the key differences between these approaches.

2. While EGNN appears as a baseline in Table 2, it is missing from the comprehensive comparison in Table 1. This makes it difficult to fully assess CDE-GNN's performance against this closely related method.

3. Figure 1 analyzes the Dirichlet energy and edge space length for GAT, but lacks a corresponding visualization showing how CDE-GNN's Dirichlet energy behaves across different layers. Adding this visualization would help demonstrate the effectiveness of the proposed method in preventing energy decay.

[1]  Zhou et al. Dirichlet energy constrained learning for deep graph neural networks. Advances in Neural Information Processing Systems, 34, 2021.

---

### Meta-Review · Area_Chair_gmW7 · 2024-12-20

**Metareview:**

This paper introduces Charge Dirichlet Energy, a geometric approach that improves deep GCN training by preserving geometric structure. Experiments show the model mitigates over-smoothing, enhances performance, and outperforms state-of-the-art methods, with strong theoretical and empirical support for the importance of geometry in GNNs.

### Strengths:

1. The problem addressed is important.
2. Empirical validation is provided across multiple datasets.

### Weaknesses:

1. The proposed method offers limited novelty.
2. The theoretical results are not new and have been previously derived in the literature.
3. The presentation requires improvement.
4. Key experimental evaluations are missing.

### Overall:

The paper exhibits significant weaknesses in terms of novelty, significance, and clarity. A rejection is therefore recommended.

**Additional Comments On Reviewer Discussion:**

The authors did not provide any feedback in the rebuttal period.

---

### Decision · Program_Chairs · 2025-01-22

Reject